# A Closer Look at the Optimization Landscapes of Generative Adversarial Networks

**Hugo Berard**[*]
Mila, Université de Montréal
Facebook AI Research

**Gauthier Gidel**[*]
Mila, Université de Montréal
Element AI

**Amjad Almahairi**
Element AI

**Pascal Vincent**[†]
Mila, Université de Montréal
Facebook AI Research

**Simon Lacoste-Julien**[†]
Mila, Université de Montréal
Element AI

## Abstract

Generative adversarial networks have been very successful in generative modeling, however they remain relatively challenging to train compared to standard deep neural networks. In this paper, we propose new visualization techniques for the optimization landscapes of GANs that enable us to study the game vector field resulting from the concatenation of the gradient of both players. Using these visualization techniques we try to bridge the gap between theory and practice by showing empirically that the training of GANs exhibits significant rotations around Local Stable Stationary Points (LSSP), similar to the one predicted by theory on toy examples. Moreover, we provide empirical evidence that GAN training converge to a *stable* stationary point which is a saddle point for the generator loss, not a minimum, while still achieving excellent performance.[1]

## 1 Introduction

Deep neural networks have exhibited remarkable success in many applications (Krizhevsky et al., 2012). This success has motivated many studies of their non-convex loss landscape (Choromanska et al., 2015; Kawaguchi, 2016; Li et al., 2018b), which, in turn, has led to many improvements, such as better initialization and optimization methods (Glorot and Bengio, 2010; Kingma and Ba, 2015).

While most of the work on studying non-convex loss landscapes has focused on single objective minimization, some recent class of models require the joint minimization of several objectives, making their optimization landscape intrinsically different. Among these models is the generative adversarial network (GAN) (Goodfellow et al., 2014) which is based on a two-player game formulation and has achieved state-of-the-art performance on some generative modeling tasks such as image generation (Brock et al., 2019).

On the theoretical side, many papers studying multi-player games have argued that one main optimization issue that arises in this case is the rotation due to the adversarial component of the game (Mescheder et al., 2018; Balduzzi et al., 2018; Gidel et al., 2019b). This has been extensively studied on toy examples, in particular on the so-called bilinear example (Goodfellow, 2016) (a.k.a Dirac GAN (Mescheder et al., 2018)). However, those toy examples are very far from the standard realistic setting of image generation involving deep networks and challenging datasets. To our knowledge it remains an open question if this rotation phenomenon actually occurs when training GANs in more practical settings.

In this paper, we aim at closing this gap between theory and practice. Following Mescheder et al. (2017) and Balduzzi et al. (2018), we argue that instead of studying the loss surface, we should study the *game vector field* (i.e., the concatenation of each player's gradient), which can provide

---

[*]Equal contributions. Correspondence to `firstname.lastname@umontreal.ca`.
[†]Canada CIFAR AI Chair (held at Mila)
[1]Code available at `https://bit.ly/2kwTu87`

better insights to the problem. To this end, we propose a new visualization technique that we call *Path-angle* which helps us observe the nature of the game vector field close to a stationary point for high dimensional models, and carry on an empirical investigation of the properties of the optimization landscape of GANs. The core questions we want to address may be summarized as the following:

> *Is rotation a phenomenon that occurs when training GANs on real world datasets, and do existing training methods find local Nash equilibria?*

To answer this question we conducted extensive experiments by training different GAN formulations (NSGAN and WGAN-GP) with different optimizers (Adam and ExtraAdam) on three datasets (MoG, MNIST and CIFAR10). Based on our experiments and using our visualization techniques we observe that the landscape of GANs is fundamentally different from the standard loss surfaces of deep networks. Furthermore, we provide evidence that existing GAN training methods do not converge to a local Nash equilibrium.

**Contributions** More precisely, our contributions are the following: (i) We propose studying empirically the game vector field (as opposed to studying the loss surfaces of each player) to understand training dynamics in GANs using a novel visualization tool, which we call *Path-angle* and that captures the rotational and attractive behaviors near local stationary points (ref. §4.2). (ii) We observe experimentally on both a mixture of Gaussians, MNIST and CIFAR10 datasets that a variety of GAN formulations have a significant rotational behavior around their locally stable stationary points (ref. §5.1). (iii) We provide empirical evidence that existing training procedures find stable stationary points that are saddle points, not minima, for the loss function of the generator (ref. § 5.2).

## 2 RELATED WORK

Improving the training of GANs has been an active research area in the past few years. Most efforts in stabilizing GAN training have focused on formulating new objectives (Arjovsky et al., 2017), or adding regularization terms (Gulrajani et al., 2017; Mescheder et al., 2017; 2018). In this work, we try to characterize the difference in the landscapes induced by different GAN formulations and how it relates to improving the training of GANs.

Recently, Nagarajan and Kolter (2017); Mescheder et al. (2018) show that a local analysis of the eigenvalues of the Jacobian of the game can provide guarantees on local stability properties. However, their theoretical analysis is based on some unrealistic assumptions such as the generator's ability to fully capture the real distribution. In this work, we assess experimentally to what extent these theoretical stability results apply in practice.

Rotations in differentiable games has been mentioned and interpreted by (Mescheder et al., 2018; Balduzzi et al., 2018) and Gidel et al. (2019b). While these papers address rotations in games from a theoretical perspective, it was never shown that GANs, which are games with highly non-convex losses, suffered from these rotations in practice. To our knowledge, trying to quantify that GANs actually suffer from this rotational component in practice for real world dataset is novel.

The stable points of the gradient dynamics in general games have been studied independently by Mazumdar and Ratliff (2018) and Adolphs et al. (2018). They notice that the locally stable stationary point of some games are not local Nash equilibria. In order to reach a local Nash equilibrium, Adolphs et al. (2018); Mazumdar et al. (2019) develop techniques based on second order information. In this work, we argue that reaching local Nash equilibria may not be as important as one may expect and that we do achieve good performance at a locally stable stationary point.

Several works have studied the loss landscape of deep neural networks. Goodfellow et al. (2015) proposed to look at the linear path between two points in parameter space and show that neural networks behave similarly to a convex loss function along this path. Draxler et al. (2018) proposed an extension where they look at nonlinear paths between two points and show that local minima are connected in deep neural networks. Another extension was proposed by (Li et al., 2018a) where they use contour plots to look at the 2D loss surface defined by two directions chosen appropriately. In this paper, we use a similar approach of following the linear path between two points to gain insight about GAN optimization landscapes. However, in this context, looking at the loss of both players along that path may be uninformative. We propose instead to look, along a linear path from initialization to best solution, at the game vector field, particularly at its angle w.r.t. the linear path, the *Path-angle*.

Another way to gain insight into the landscape of deep neural networks is by looking at the Hessian of the loss; this was done in the context of single objective minimization by (Dauphin et al., 2014; Sagun et al., 2016; 2017; Alain et al., 2019). Compared to linear path visualizations which can give global information (but only along one direction), the Hessian provides information about the loss landscape in several directions but only locally. The full Hessian is expensive to compute and one often has to resort to approximations such has computing only the top-k eigenvalues. While, the Hessian is symmetric and thus has real eigenvalues, the Jacobian of a game vector field is significantly different since it is in general not symmetric, which means that the eigenvalues belong to the complex plane. In the context of GANs, Mescheder et al. (2017) introduced a gradient penalty and use the eigenvalues of the Jacobian of the game vector field to show its benefits in terms of stability. In our work, we compute these eigenvalues to assess that, on different GAN formulations and datasets, existing training procedures find a locally stable stationary point that is a saddle point for the loss function of the generator.

## 3 FORMULATIONS FOR GAN OPTIMIZATION AND THEIR PRACTICAL IMPLICATIONS

### 3.1 THE STANDARD GAME THEORY FORMULATION

From a game theory point of view, GAN training may be seen as a game between two players: the discriminator $D_{\boldsymbol{\varphi}}$ and the generator $G_{\boldsymbol{\theta}}$, each of which is trying to minimize its loss $\mathcal{L}_D$ and $\mathcal{L}_G$, respectively. Using the same formulation as Mescheder et al. (2017), the GAN objective takes the following form (for simplicity of presentation, we focus on the unconstrained formulation):

$$\boldsymbol{\theta}^* \in \underset{\boldsymbol{\theta} \in \mathbb{R}^p}{\arg\min} \, \mathcal{L}_G(\boldsymbol{\theta}, \boldsymbol{\varphi}^*) \quad \text{and} \quad \boldsymbol{\varphi}^* \in \underset{\boldsymbol{\varphi} \in \mathbb{R}^d}{\arg\min} \, \mathcal{L}_D(\boldsymbol{\theta}^*, \boldsymbol{\varphi}) \,. \tag{1}$$

The solution $(\boldsymbol{\theta}^*, \boldsymbol{\varphi}^*)$ is called a *Nash equilibrium* (NE). In practice, the considered objectives are non-convex and we typically cannot expect better than a *local* Nash equilibrium (LNE), i.e. a point at which (1) is only locally true (see e.g. (Adolphs et al., 2018) for a formal definition). Ratliff et al. (2016) derived some derivative-based necessary and sufficient conditions for being a LNE. They show that, for being a local NE it is sufficient to be a *differential Nash equilibrium*:

**Definition 1** (Differential NE). *A point* $(\boldsymbol{\theta}^*, \boldsymbol{\varphi}^*)$ *is a* differential Nash equilibrium *(DNE) iff*

$$\|\nabla_{\boldsymbol{\theta}} \mathcal{L}_G(\boldsymbol{\theta}^*, \boldsymbol{\varphi}^*)\| = \|\nabla_{\boldsymbol{\varphi}} \mathcal{L}_D(\boldsymbol{\theta}^*, \boldsymbol{\varphi}^*)\| = 0, \, \nabla_{\boldsymbol{\theta}}^2 \mathcal{L}_G(\boldsymbol{\theta}^*, \boldsymbol{\varphi}^*) \succ 0 \text{ and } \nabla_{\boldsymbol{\varphi}}^2 \mathcal{L}_D(\boldsymbol{\theta}^*, \boldsymbol{\varphi}^*) \succ 0 \quad (2)$$

*where* $\boldsymbol{S} \succ 0$ *if and only if* $\boldsymbol{S}$ *is positive definite.*

Being a DNE is not necessary for being a LNE because a local Nash equilibrium may have Hessians that are only semi-definite. NE are commonly used in GANs to describe the goal of the learning procedure (Goodfellow et al., 2014): in this definition, $\boldsymbol{\theta}^*$ (resp. $\boldsymbol{\varphi}^*$) is seen as a local minimizer of $\mathcal{L}_G(\cdot, \boldsymbol{\varphi}^*)$ (resp. $\mathcal{L}_D(\boldsymbol{\theta}^*, \cdot)$).

Under this view, however, the interaction between the two networks is not taken into account. This is an important aspect of the game stability that is missed in the definition of DNE (and Nash equilibrium in general). We illustrate this point in the following section, where we develop an example of a game for which gradient methods converge to a point which is a saddle point for the generator's loss and thus not a DNE for the game.

### 3.2 AN ALTERNATIVE FORMULATION BASED ON THE GAME VECTOR FIELD

In practice, GANs are trained using first order methods that compute the gradients of the losses of each player. Following Gidel et al. (2019a), an alternative point of view on optimizing GANs is to jointly consider the players' parameters $\boldsymbol{\theta}$ and $\boldsymbol{\varphi}$ as a joint state $\boldsymbol{\omega} := (\boldsymbol{\theta}, \boldsymbol{\varphi})$, and to study the vector field associated with these gradients,[2] which we call the *game vector field*

$$\boldsymbol{v}(\boldsymbol{\omega}) := \begin{bmatrix} \nabla_{\boldsymbol{\theta}} \mathcal{L}_G(\boldsymbol{\omega})^\top & \nabla_{\boldsymbol{\varphi}} \mathcal{L}_D(\boldsymbol{\omega})^\top \end{bmatrix}^\top \quad \text{where} \quad \boldsymbol{\omega} := (\boldsymbol{\theta}, \boldsymbol{\varphi}) \,. \tag{3}$$

---

[2]Note that, in practice, the joint vector field (3) is *not* a gradient vector field, i.e., it cannot be rewritten as the gradient of a single function.

| Zero-sum game | Non-zero-sum game |
|---|---|
| NE $\Rightarrow$ LSSE (Mescheder et al., 2018) | NE $\not\Rightarrow$ LSSE (Example 2, §A.2) |
| NE $\not\Leftarrow$ LSSE (Adolphs et al., 2018) | NE $\not\Leftarrow$ LSSE (Example 1) |

Table 1: Summary of the implications between Differentiable Nash Equilibrium (DNE) and a locally stable stationnary point (LSSP): in general, being a DNE is neither necessary or sufficient for being a LSSP.

With this perspective, the notion of DNE is replaced by the notion of locally stable stationary point (LSSP). Verhulst (1989, Theorem 7.1) defines a LSSP $\boldsymbol{\omega}^*$ using the eigenvalues of the Jacobian of the game vector field $\nabla \boldsymbol{v}(\boldsymbol{\omega}^*)$ at that point.

**Definition 2** (LSSP). *A point $\boldsymbol{\omega}^*$ is a* locally stable stationary point *(LSSP) iff*

$$\boldsymbol{v}(\boldsymbol{\omega}^*) = 0 \qquad and \qquad \Re(\lambda) > 0 \,, \quad \forall \lambda \in \mathrm{Sp}(\nabla \boldsymbol{v}(\boldsymbol{\omega}^*)) \,. \tag{4}$$

*where $\Re$ denote the real part of the eigenvalue $\lambda$ belonging to the spectrum of $\nabla \boldsymbol{v}(\boldsymbol{\omega}^*)$.*

This definition is not easy to interpret but one can intuitively understand a LSSP as a stationary point (a point $\boldsymbol{\omega}^*$ where $\boldsymbol{v}(\boldsymbol{\omega}^*) = 0$) to which all neighbouring points are attracted. We will formalize this intuition of attraction in Proposition 1. In our two-player game setting, the Jacobian of the game vector field around the LSSP has the following block-matrices form:

$$\nabla \boldsymbol{v}(\boldsymbol{\omega}^*) = \begin{bmatrix} \nabla_{\boldsymbol{\theta}}^2 \mathcal{L}_G(\boldsymbol{\omega}^*) & \nabla_{\boldsymbol{\varphi}} \nabla_{\boldsymbol{\theta}} \mathcal{L}_G(\boldsymbol{\omega}^*) \\ \nabla_{\boldsymbol{\theta}} \nabla_{\boldsymbol{\varphi}} \mathcal{L}_D(\boldsymbol{\omega}^*) & \nabla_{\boldsymbol{\varphi}}^2 \mathcal{L}_D(\boldsymbol{\omega}^*) \end{bmatrix} = \begin{bmatrix} \boldsymbol{S}_1 & \boldsymbol{B} \\ \boldsymbol{A} & \boldsymbol{S}_2 \end{bmatrix} \,. \tag{5}$$

When $\boldsymbol{B} = -\boldsymbol{A}^\top$, being a DNE is a sufficient condition for being of LSSP (Mazumdar and Ratliff, 2018). However, some LSSP may not be DNE (Adolphs et al., 2018), meaning that the optimal generator $\boldsymbol{\theta}^*$ could be a saddle point of $\mathcal{L}_G(\cdot, \boldsymbol{\varphi}^*)$, while the optimal joint state $(\boldsymbol{\theta}^*, \boldsymbol{\varphi}^*)$ may be a LSSP of the game. We summarize these properties in Table 1. In order to illustrate the intuition behind this counter-intuitive fact, we study a simple example where the generator is 2D and the discriminator is 1D.

**Example 1.** *Let us consider $\mathcal{L}_G$ as a hyperbolic paraboloid (a.k.a., saddle point function) centered in $(1, 1)$ where $(1, \varphi)$ is the principal descent direction and $(-\varphi, 1)$ is the principal ascent direction, while $\mathcal{L}_D$ is a simple bilinear objective.*

$$\mathcal{L}_G(\theta_1, \theta_2, \varphi) = (\theta_2 - \varphi\theta_1 - 1)^2 - \tfrac{1}{2}(\theta_1 + \varphi\theta_2 - 1)^2 \,, \quad \mathcal{L}_D(\theta_1, \theta_2, \varphi) = \varphi(5\theta_1 + 4\theta_2 - 9)$$

*We plot $\mathcal{L}_G$ in Fig. 1b. Note that the discriminator $\varphi$ controls the principal descent direction of $\mathcal{L}_G$.*

We show (see § A.2) that $(\theta_1^*, \theta_2^*, \varphi^*) = (1, 1, 0)$ is a locally stable stationary point but is not a DNE: the generator loss at the optimum $(\theta_1, \theta_2) \mapsto \mathcal{L}_G(\theta_1, \theta_2, \varphi^*) = \theta_2^2 - \tfrac{1}{2}\theta_1^2$ is not at a DNE because it has a clear descent direction, $(1, 0)$. However, if the generator follows this descent direction, the dynamics will remain stable because the discriminator will update its parameter, rotating the saddle and making $(1, 0)$ an ascent direction. We call this phenomenon *dynamic stability*: the loss $\mathcal{L}_G(\cdot, \varphi^*)$ is unstable for a fixed $\varphi^*$ but becomes stable when $\varphi$ dynamically interacts with the generator around $\varphi^*$.

A mechanical analogy for this dynamic stability phenomenon is a ball in a rotating saddle—even though the gravity pushes the ball to escape the saddle, a quick enough rotation of the saddle would trap the ball at the center (see (Thompson et al., 2002) for more details). This analogy has been used to explain Paul's trap (Paul, 1990): a counter-intuitive way to trap ions using a dynamic electric field. In Example 1, the parameter $\varphi$ explicitly controls the rotation of the saddle.

This example illustrates the fact that the DNE corresponds to a notion of *static stability*: it is the stability of one player's loss given the other player is fixed. Conversely, LSSP captures a notion of *dynamic stability* that considers both players jointly.

By looking at the game vector field we capture these interactions. Fig. 1b only captures a snapshot of the generator's loss surface for a fixed $\varphi$ and indicates static instability (the generator is at a saddle point of its loss). In Fig. 1a, however, one can see that, starting from any point, we will rotate around the stationary point $(\varphi^*, \theta_1^*) = (0, 1)$ and eventually converge to it.

The visualization of the game vector field reveals an interesting behavior that does not occur in single objective minimization: close to a LSSP, the parameters rotate around it. Understanding this phenomenon is key to grasp the optimization difficulties arising in games. In the next section, we

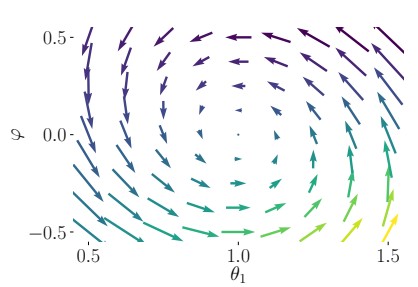

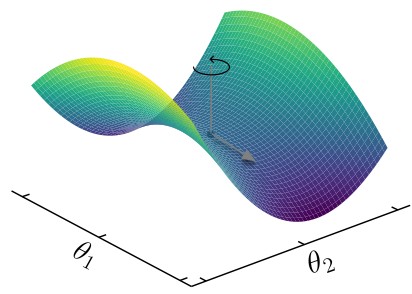

(a) 2D projection of the vector field.                    (b) Landscape of the generator loss.

Figure 1: Visualizations of Example 1. Left: projection of the game vector field on the plane $\theta_2 = 1$. Right: Generator loss. The descent direction is $(1, \varphi)$ (in grey). As the generator follows this descent direction, the discriminator changes the value of $\varphi$, making the saddle rotate, as indicated by the circular black arrow.

formally characterize the notion of rotation around a LSSP and in §4 we develop tools to visualize it in high dimensions. Note that gradient methods may converge to saddle points in single objective minimization, but these are not *stable* stationary points, unlike in our game example.

### 3.3    ROTATION AND ATTRACTION AROUND LOCALLY STABLE STATIONARY POINTS IN GAMES

In this section, we formalize the notions of rotation and attraction around LSSP in games, which we believe may explain some difficulties in GAN training. The local stability of a LSSP is characterized by the eigenvalues of the Jacobian $\nabla v(\omega^*)$ because we can linearize $v(\omega)$ around $\omega^*$:

$$v(\omega) \approx \nabla v(\omega^*)(\omega - \omega^*). \tag{6}$$

If we assume that (6) is an equality, we have the following theorem.

**Proposition 1.** *Let us assume that* (6) *is an equality and that* $\nabla v(\omega^*)$ *is diagonalizable, then there exists a basis* $P$ *such that the coordinates* $\tilde{\omega}_j(t) := [P(\omega(t) - \omega^*)]_j$ *where* $\omega(t)$ *is a solution of* (6) *have the following behavior: for* $\lambda_j \in \mathrm{Sp}\,\nabla v(\omega^*)$ *we have,*

*1. If* $\lambda_j \in \mathbb{R}$, *we observe* pure attraction*:*    $\tilde{\omega}_j(t) = e^{-\lambda_j t}\tilde{\omega}_j(0)$.

*2. If* $\Re(\lambda_j) = 0$, *we observe* pure rotation*:* $\begin{bmatrix} \tilde{\omega}_j(t) \\ \tilde{\omega}_{j+1}(t) \end{bmatrix} = \begin{bmatrix} \cos|\lambda_j t| & \sin|\lambda_j t| \\ -\sin|\lambda_j t| & \cos|\lambda_j t| \end{bmatrix} \begin{bmatrix} \tilde{\omega}_j(0) \\ \tilde{\omega}_{j+1}(0) \end{bmatrix}$.

*3. Otherwise, we observe both:* $\begin{bmatrix} \tilde{\omega}_j(t) \\ \tilde{\omega}_{j+1}(t) \end{bmatrix} = e^{-\mathrm{Re}(\lambda_j)t} \begin{bmatrix} \cos\mathrm{Im}(\lambda_j t) & \sin\mathrm{Im}(\lambda_j t) \\ -\sin\mathrm{Im}(\lambda_j t) & \cos\mathrm{Im}(\lambda_j t) \end{bmatrix} \begin{bmatrix} \tilde{\omega}_j(0) \\ \tilde{\omega}_{j+1}(0) \end{bmatrix}$.

*Note that we re-ordered the eigenvalues such that the complex conjugate eigenvalues form pairs: if* $\lambda_j \notin \mathbb{R}$ *then* $\lambda_{j+1} = \bar{\lambda}_j$.

Matrices in 2. and 3. are rotations matrices. They induce a rotational behavior illustrated in Fig 1a.

This proposition shows that the dynamics of $\omega(t)$ can be decomposed in a particular basis into attractions and rotations over components that do not interact between each other. Rotation does not appear in single objective minimization around a local minimum, because the eigenvalues of the Hessian of the objective are always real. Mescheder et al. (2017) discussed that difficulties in training GANs may be a result of the imaginary part of the eigenvalues of the Jacobian of the game vector field and Gidel et al. (2019b) mentioned that games have a natural oscillatory behavior. This cyclic behavior has been explained in (Balduzzi et al., 2018) by a non-zero Hamiltonian component in the Helmholtz decomposition of the Jacobian of the game vector field. All these explanations are related to the spectral properties of this Jacobian. The goal of Proposition 1 is to provide a formal definition to the notions of *rotation* and *attraction* we are dealing with in this paper.

In the following section, we introduce a new tool in order to assess the magnitude of the rotation around a LSSP compared to the attraction to this point.

## 4 VISUALIZATION FOR THE VECTOR FIELD LANDSCAPE

Neural networks are parametrized by a large number of variables and visualizations are only possible using low dimensional plots (1D or 2D). We first present a standard visualization tool for deep neural network loss surfaces that we will exploit in §4.2.

### 4.1 STANDARD VISUALIZATIONS FOR THE LOSS SURFACE

One way to visualize a neural network's loss landscape is to follow a parametrized path $\boldsymbol{\omega}(\alpha)$ that connects two parameters $\boldsymbol{\omega}, \boldsymbol{\omega}'$ (often one is chosen early in learning and another one is chosen late in learning, close to a solution). A path is a continuous function $\boldsymbol{\omega}(\cdot)$ such that $\boldsymbol{\omega}(0) = \boldsymbol{\omega}$ and $\boldsymbol{\omega}(1) = \boldsymbol{\omega}'$. Goodfellow et al. (2015) considered a linear path $\boldsymbol{\omega}(\alpha) = \alpha\boldsymbol{\omega} + (1-\alpha)\boldsymbol{\omega}'$. More complex paths can be considered to assess whether different minima are connected (Draxler et al., 2018).

### 4.2 PROPOSED VISUALIZATION: PATH-ANGLE

We propose to study the linear path between parameters early in learning and parameters late in learning. We illustrate the extreme cases for the game vector field along this path in simple examples in Figure 2(a-c): pure attraction occurs when the vector field perfectly points to the optimum (Fig. 2a) and pure rotation when the vector field is orthogonal to the direction to the optimum (Fig. 2b). In practice, we expect the vector field to be in between these two extreme cases (Fig. 2c). In order to determine in which case we are, around a LSSP, in practice, we propose the following tools.

**Path-norm.** We first ensure that we are in a neighborhood of a stationary point by computing the norm of the vector field. Note that considering independently the norm of each player may be misleading: even though the gradient of one player may be close to zero, it does not mean that we are at a stationary point since the other player might still be updating its parameters.

**Path-angle.** Once we are close to a final point $\boldsymbol{\omega}'$, i.e., in a neighborhood of a LSSP, we propose to look at the angle between the vector field (3) and the linear path from $\boldsymbol{\omega}$ to $\boldsymbol{\omega}'$. Specifically, we monitor the cosine of this angle, a quantity we call *Path-angle*:

$$c(\alpha) := \frac{\langle \boldsymbol{\omega}' - \boldsymbol{\omega}, \boldsymbol{v}_\alpha \rangle}{\|\boldsymbol{\omega}' - \boldsymbol{\omega}\|\|\boldsymbol{v}_\alpha\|} \quad \text{where} \quad \boldsymbol{v}_\alpha := \boldsymbol{v}(\alpha\boldsymbol{\omega}' + (1-\alpha)\boldsymbol{\omega}), \; \alpha \in [a, b]. \tag{7}$$

Usually $[a, b] = [0, 1]$, but since we are interested in the landscape around a LSSP, it might be more informative to also consider further extrapolated points around $\boldsymbol{\omega}'$ with $b > 1$.

**Eigenvalues of the Jacobian.** Another important tool to gain insights on the behavior close to a LSSP, as discussed in §3.2, is to look at the eigenvalues of $\nabla\boldsymbol{v}(\boldsymbol{\omega}^*)$. We propose to compute the top-k eigenvalues of this Jacobian. When all the eigenvalues have positive real parts, we conclude that we have reached a LSSP, and if some eigenvalues have large imaginary parts, then the game has a strong rotational behavior (Thm. 1). Similarly, we can also compute the top-k eigenvalues of the diagonal blocks of the Jacobian, which correspond to the Hessian of each player. These eigenvalues can inform us on whether we have converged to a LSSP that is not a LNE.

An important advantage of the Path-angle relative to the computation of the eigenvalues of $\nabla\boldsymbol{v}(\boldsymbol{\omega}^*)$ is that it only requires computing gradients (and not second order derivatives, which may be prohibitively computationally expensive for deep networks). Also, it provides information along a whole path between two points and thus, more global information than the Jacobian computed at a single point. In the following section, we use the Path-angle to study the archetypal behaviors presented in Thm 1.

### 4.3 ARCHETYPAL BEHAVIORS OF THE PATH-ANGLE AROUND A LSSP

Around a LSSP, we have seen in (6) that the behavior of the vector field is mainly dictated by the Jacobian matrix $\nabla\boldsymbol{v}(\boldsymbol{\omega}^*)$. This motivates the study of the behavior of the Path-angle $c(\alpha)$ where the Jacobian is a constant matrix:

$$\boldsymbol{v}(\boldsymbol{\omega}) = \begin{bmatrix} \boldsymbol{S}_1 & \boldsymbol{B} \\ \boldsymbol{A} & \boldsymbol{S}_2 \end{bmatrix}(\boldsymbol{\omega} - \boldsymbol{\omega}^*) \quad \text{and thus} \quad \nabla\boldsymbol{v}(\boldsymbol{\omega}) = \begin{bmatrix} \boldsymbol{S}_1 & \boldsymbol{B} \\ \boldsymbol{A} & \boldsymbol{S}_2 \end{bmatrix} \quad \forall\boldsymbol{\omega}. \tag{8}$$

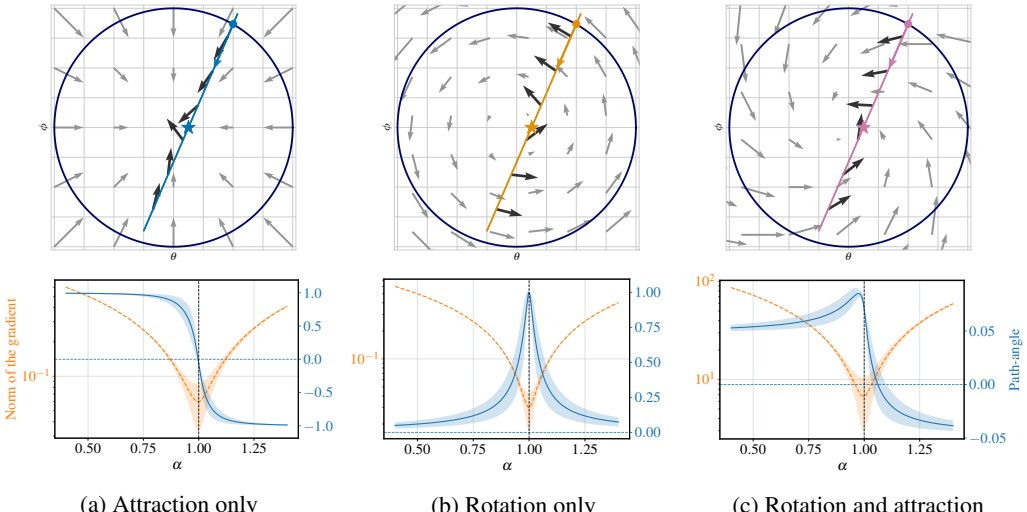

(a) Attraction only      (b) Rotation only      (c) Rotation and attraction

Figure 2: **Above**: game vector field (in grey) for different archetypal behaviors. The equilibrium of the game is at $(0,0)$. Black arrows correspond to the directions of the vector field at different linear interpolations between two points: ● and ★. **Below**: path-angle $c(\alpha)$ for different archetypal behaviors (right y-axis, in blue). The left y-axis in orange correspond to the norm of the gradients. Notice the "bump" in path-angle (close to $\alpha = 1$), characteristic of rotational dynamics.

Depending on the choice of $\boldsymbol{S}_1, \boldsymbol{S}_2, \boldsymbol{A}$ and $\boldsymbol{B}$, we cover the following cases:

- $\boldsymbol{S}_1, \boldsymbol{S}_2 \succ 0, \boldsymbol{A} = \boldsymbol{B} = 0$: eigenvalues are real. Thm. 1 ensures that we only have *attraction*. Far from $\boldsymbol{\omega}^*$, the gradient points to $\boldsymbol{\omega}^*$ (See Fig. 2a) and thus $c(\alpha) = 1$ for $\alpha \ll 1$ and $c(\alpha) = -1$ for $\alpha \gg 1$. Since $\boldsymbol{\omega}'$ is not exactly $\boldsymbol{\omega}^*$, we observe a *quick sign switch* of the Path-angle around $\alpha = 1$. We plotted the average Path-angle over different approximate optima in Fig. 2a (see appendix for details).

- $\boldsymbol{S}_1, \boldsymbol{S}_2 = 0, \boldsymbol{A} = -\boldsymbol{B}^\top$: eigenvalues are pure imaginary. Thm. 1 ensures that we only have *rotations*. Far from the optimum the gradient is orthogonal to the direction that points to $\boldsymbol{\omega}$ (See Fig. 2b). Thus, $c(\alpha)$ vanishes for $\alpha \ll 1$ and $\alpha \gg 1$. Because $\boldsymbol{\omega}'$ is not exactly $\boldsymbol{\omega}^*$, around $\alpha = 1$, the gradient is tangent to the circles induced by the rotational dynamics and thus $c(\alpha) = \pm 1$. That is why in Fig. 2b we observe a *bump* in $c(\alpha)$ when $\alpha$ is close to 1.

- General high dimensional LSSP (4). The dynamics display both attraction and rotation. We observe a combination of the sign switch due to the attraction and the bump due to the rotation. The higher the bump, the closer we are to pure rotations. Since we are performing a low dimensional visualization, we actually project the gradient onto our direction of interest. That is why the Path-angle is significantly smaller than 1 in Fig. 2c.

## 5 NUMERICAL RESULTS ON GANS

**Losses.** We focus on two common GAN loss formulations: we consider both the original non-saturating GAN (NSGAN) formulation proposed in Goodfellow et al. (2014) and the WGAN-GP objective described in Gulrajani et al. (2017).

**Datasets.** We first propose to train a GAN on a toy task composed of a 1D mixture of 2 Gaussians (MoG) with 10,000 samples. For this task both the generator and discriminator are neural networks with 1 hidden layer and ReLU activations. We also train a GAN on MNIST, where we use the DCGAN architecture (Radford et al., 2016) with spectral normalization(see §C.2 for details). Finally we also look at the optimization landscape of a state of the art ResNet on CIFAR10 (Krizhevsky and Hinton, 2009).

**Optimization methods.** For the mixture of Gaussian (MoG) dataset, we used the full-batch extragradient method (Korpelevich, 1976; Gidel et al., 2019a). We also tried to use standard batch gradient descent, but this led to unstable results indicating that gradient descent might indeed be unable to

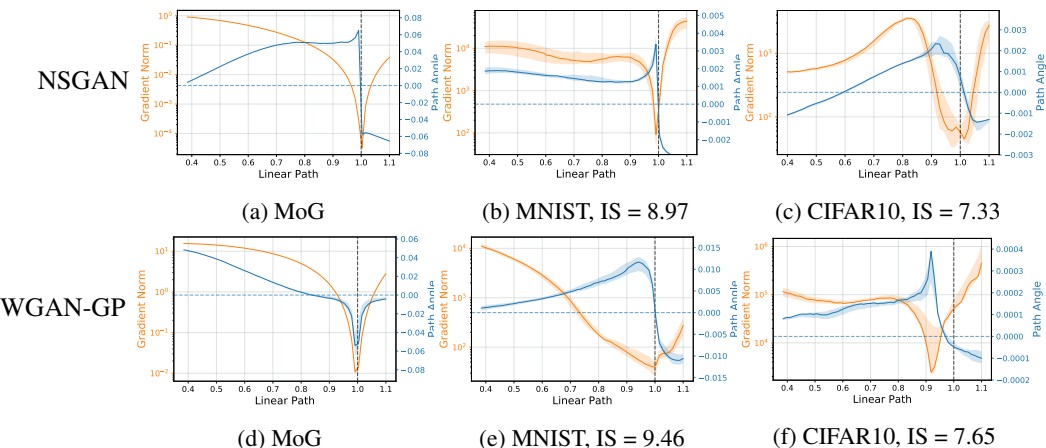

Figure 3: Path-angle for NSGAN (**top row**) and WGAN-GP (**bottom row**) trained on the different datasets, see Appendix C.3 for details on how the path-angle is computed. For MoG the ending point is a generator which has learned the distribution. For MNIST and CIFAR10 we indicate the Inception score (IS) at the ending point of the interpolation. Notice the "bump" in path-angle (close to $\alpha = 1.0$), characteristic of games rotational dynamics, and absent in the minimization problem (d). Details on error bars in §C.3.

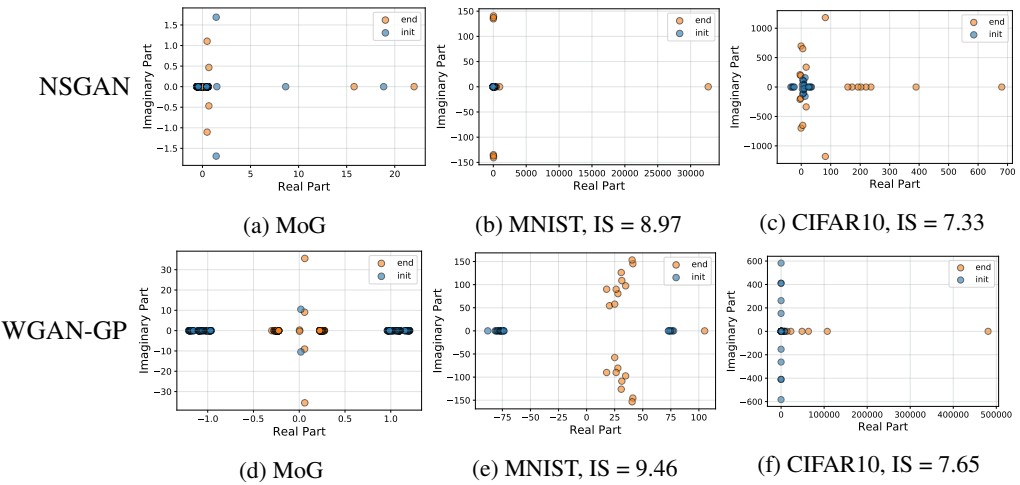

Figure 4: Eigenvalues of the Jacobian of the game for NSGAN (**top row**) and WGAN-GP (**bottom row**) trained on the different datasets. Large imaginary eigenvalues are characteristic of rotational behavior. Notice that NSGAN and WGAN-GP objectives lead to very different landscapes (see how the eigenvalues of WGAN-GP are shifted to the right of the imaginary axis). This could explain the difference in performance between NSGAN and WGAN-GP.

converge to stable stationary points due to the rotations (see §C.4). On MNIST and CIFAR10, we tested both Adam (Kingma and Ba, 2015) and ExtraAdam (Gidel et al., 2019a). The observations made on models trained with both methods are very similar. ExtraAdam gives slightly better performance in terms of inception score (Salimans et al., 2016), and Adam sometimes converge to unstable points, thus we decided to only include the observations on ExtraAdam, for more details on the observations on Adam (see §C.5). As recommended by Heusel et al. (2017), we chose different learning rates for the discriminator and the generator. All the hyper-parameters and precise details about the experiments can be found in §C.1.

## 5.1 EVIDENCE OF ROTATION AROUND LOCALLY STABLE STATIONARY POINTS IN GANS

We first look, for all the different models and datasets, at the path-angles between a random initialization (initial point) and the set of parameters during training achieving the best performance (end point) (Fig. 3), and at the eigenvalues of the Jacobian of the game vector field for the same end point (Fig. 4). We're mostly interested in looking at the optimization landscape around LSSPs, so we first check if we are actually close to one. To do so we look at the gradient norm around the end point, this is shown by the orange curves in Fig.3, we can see that the norm of the gradient is quite small for all the models meaning that we are close to a stationary point. We also need to check that the point is stable, to do so we look at the eigenvalues of the Game in Fig. 4, if all the eigenvalues have positive real parts then the point is also stable. We observe that most of the time, the model has reached a LSSP. However we can see that this is not always the case, for example in Fig. 4d some of the eigenvalues have a negative real part. We still include those results since although the point is unstable it gives similar performance to a LSSP.

Our first observation is that all the GAN objectives on both datasets have a non zero rotational component. This can be seen by looking at the Path-angle in Fig. 3, where we always observe a bump, and this is also confirmed by the large imaginary part in the eigenvalues of the Jacobian in Fig. 4. The rotational component is clearly visible in Fig. 3d, where we see no sign switch and a clear bump similar to Fig. 2b. On MNIST and CIFAR10, with NSGAN and WGAN-GP (see Fig. 3), we observe a combination of a bump and a sign switch similar to Fig. 2c. Also Fig. 4 clearly shows the existence of imaginary eigenvalues with large magnitude. Fig. 4c and 4e. We can see that while almost all models exhibit rotations, the distribution of the eigenvalues are very different. In particular the complex eigenvalues for NSGAN seems to be much more concentrated on the imaginary axis while WGAN-GP tends to spread the eigenvalues towards the right of the imaginary axis Fig. 4e. This shows that different GAN objectives can lead to very different landscapes, and has implications in terms of optimization, in particular that might explain why WGAN-GP performs slightly better than NSGAN.

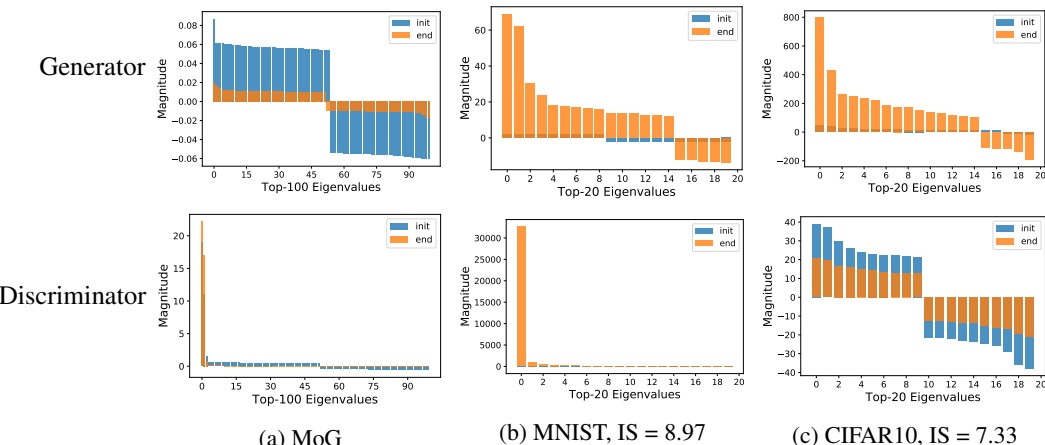

(a) MoG      (b) MNIST, IS = 8.97      (c) CIFAR10, IS = 7.33

Figure 5: **NSGAN.** Top $k$-Eigenvalues of the Hessian of each player (in terms of magnitude) in descending order. Top Eigenvalues indicate that the Generator does not reach a local minimum but a saddle point (for CIFAR10 actually both the generator and discriminator are at saddle points). Thus the training algorithms converge to LSSPs which are not Nash equilibria.

## 5.2 THE LOCALLY STABLE STATIONARY POINTS OF GANS ARE NOT LOCAL NASH EQUILIBRIA

As mentioned at the beginning of §5.1, the points we are considering are most of the times LSSP. To check if these points are also local Nash equilibria (LNE) we compute the eigenvalues of the Hessian of each player independently. If all the eigenvalues of each player are positive, it means that we have reached a DNE. Since the computation of the full spectrum of the Hessians is expensive, we restrict ourselves to the top-k eigenvalues with largest magnitude: exhibiting one significant negative eigenvalue is enough to indicate that the point considered is not in the neighborhood of a

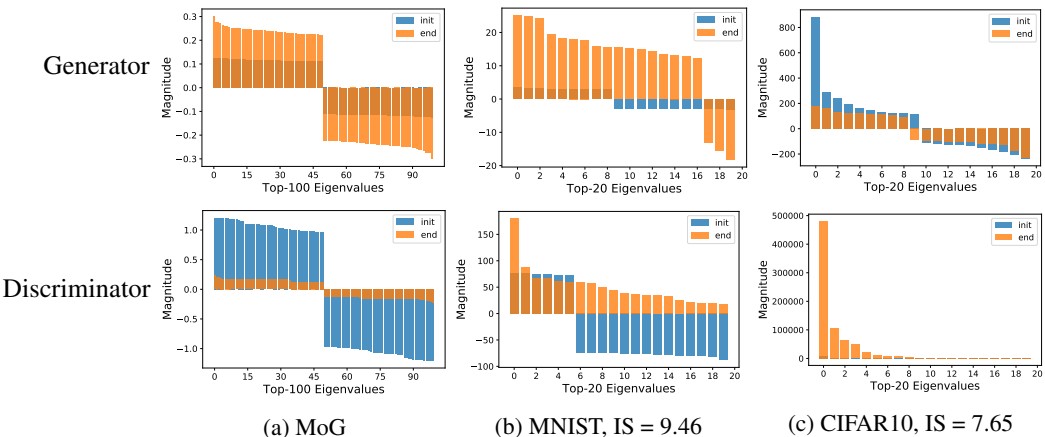

Figure 6: **WGAN-GP.** Top $k$-Eigenvalues of the Hessian of each player (in terms of magnitude) in descending order. Top Eigenvalues indicate that the Generator does not reach a local minimum but a saddle point. Thus the training algorithms converge to LSSPs which are not Nash equilibria.

LNE. Results are shown in Fig. 5 and Fig. 6, from which we make several observations. First, we see that the generator never reaches a local minimum but instead finds a saddle point. This means that the algorithm converges to a LSSP which is not a LNE, while achieving good results with respect to our evaluation metrics. This raises the question whether convergence to a LNE is actually needed or if converging to a LSSP is sufficient to reach a good solution. We also observe a large difference in the eigenvalues of the discriminator when using the WGAN-GP v.s. the NSGAN objective. In particular, we find that the discriminator in NSGAN converges to a solution with very large positive eigenvalues compared to WGAN-GP. This shows that the discriminator in NSGAN converges to a much sharper minimum. This is consistent with the fact that the gradient penalty acts as a regularizer on the discriminator and prevents it from becoming too sharp.

## 6 DISCUSSION

Across different GAN formulations, standard optimization methods and datasets, we consistently observed that GANs do not converge to local Nash equilibria. Instead the generator often ends up being at a saddle point of the generator loss function. However, in practice, these LSSP achieve really good generator performance metrics, which leads us to question whether we need a Nash equilibrium to get a generator with good performance in GANs and whether such DNE with good performance does actually exist. Moreover, we have provided evidence that the optimization landscapes of GANs typically have rotational components specific to games. We argue that these rotational components are part of the reason why GANs are challenging to train, in particular that the instabilities observed during training may come from such rotations close to LSSP. It shows that simple low dimensional examples, such as for instance Dirac GAN, does capture some of the arising challenges for training large scale GANs, thus, motivating the practical use of method able to handle strong rotational components, such as extragradient (Gidel et al., 2019a), averaging (Yazıcı et al., 2019), optimism (Daskalakis et al., 2018) or gradient penalty based methods (Mescheder et al., 2017; Gulrajani et al., 2017).

ACKNOWLEDGMENTS.

The contribution to this research by Mila, Université de Montréal authors was partially supported by the Canada CIFAR AI Chair Program (held at Mila), the Canada Excellence Research Chair in "Data Science for Realtime Decision-making", by the NSERC Discovery Grant RGPIN-2017-06936 (held at Université de Montréal), by a Borealis AI fellowship and by a Google Focused Research award. The authors would like to thank Tatjana Chavdarova for fruitful discussions.

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

# A   PROOF OF THEOREMS AND PROPOSITIONS

## A.1   PROOF OF THEOREM 1

Let us recall the theorem of interest:

**Proposition' 1.** *Let us assume that* (6) *is an equality and that* $\nabla v(\omega^*)$ *is diagonalizable, then there exists a basis* $P$ *such that the coordinates* $\tilde{\omega}(t) := P(\omega(t) - \omega^*)$ *have the following behavior,*

*1. For* $\lambda_j \in \mathrm{Sp} \, \nabla v(\omega^*)$, $\lambda_j \in \mathbb{R}$, *we observe pure attraction:*    $\tilde{\omega}_j(t) = e^{-\lambda_j t}[\tilde{\omega}_j(0)$.

*2. For* $\lambda_j \in \mathrm{Sp} \, \nabla v(\omega^*)$, $\Re(\lambda_j) = 0$, *we observe pure rotation:* $\begin{bmatrix} \tilde{\omega}_j(t) \\ \tilde{\omega}_{j+1}(t) \end{bmatrix} = R_{|\lambda_j|t} \begin{bmatrix} \tilde{\omega}_j(0) \\ \tilde{\omega}_{j+1}(0) \end{bmatrix}$.

*3. Otherwise, we observe both:* $\begin{bmatrix} \tilde{\omega}_j(t) \\ \tilde{\omega}_{j+1}(t) \end{bmatrix} = e^{-\mathrm{Re}(\lambda_j)t} R_{\mathrm{Im}(\lambda_j)t} \begin{bmatrix} \tilde{\omega}_j(0) \\ \tilde{\omega}_{j+1}(0) \end{bmatrix}$.

*The matrix* $R_\varphi$ *corresponds to a rotation of angle* $\varphi$. *Note that, we re-ordered the eigenvalues such that the complex conjugate eigenvalues form pairs: if* $\lambda_j \notin \mathbb{R}$ *then* $\lambda_{j+1} = \bar{\lambda}_j$.

*Proof.* The ODE we consider is,

$$\frac{d\omega(t)}{dt} = \nabla v(\omega^*)(\omega(t) - \omega^*) \tag{9}$$

The solution of this ODE is

$$\omega(t) = e^{-(t-t_0)\nabla v(\omega^*)}(\omega(t_0) - \omega^*) + \omega^* \tag{10}$$

Let us now consider $\lambda$ an eigenvalue of $\mathrm{Sp}(\nabla v(\omega^*))$ such that $\mathrm{Re}(\lambda) > 0$ and $\mathrm{Im}(\lambda) \neq 0$. Since $\nabla v(\omega^*)$ is a real matrix and $\mathrm{Im}(\lambda) \neq 0$ we know that the complex conjugate $\bar{\lambda}$ of $\lambda$ belongs to $\mathrm{Sp}(\nabla v(\omega^*))$. Let $u_0$ be a complex eigenvector of $\lambda$, then we have that,

$$\nabla v(\omega^*)u_0 = \lambda u_0 \quad \Rightarrow \quad \nabla v(\omega^*)\bar{u}_0 = \bar{\lambda}\bar{u}_0 \tag{11}$$

and thus $\bar{u}_0$ is a eigenvector of $\bar{\lambda}$. Now if we set $u_1 := u_0 + \bar{u}_0$ and $iu_2 := u_0 - \bar{u}_0$, we have that

$$e^{-t\nabla v(\omega^*)}u_1 = e^{-t\lambda}u_0 + e^{-t\bar{\lambda}}\bar{u}_0 = \mathrm{Re}(e^{-t\lambda})u_1 + \mathrm{Im}(e^{-t\lambda})u_2 \tag{12}$$

$$e^{-t\nabla v(\omega^*)}iu_2 = e^{-t\lambda}u_0 - e^{-t\bar{\lambda}}\bar{u}_0 = i(\mathrm{Re}(e^{-t\lambda})u_2 - \mathrm{Im}(e^{-t\lambda})u_1) \tag{13}$$

Thus if we consider the basis that diagonalizes $\nabla v(\omega^*)$ and modify the complex conjugate eigenvalues in the way we described right after 11 we get the expected diagonal form in a real basis. Thus there exists $P$ such that

$$\nabla v(\omega^*) = PDP^{-1} \tag{14}$$

where $D$ is the block diagonal matrix with the block described in Theorem 1.   □

## A.2   BEING A DNE IS NEITHER NECESSARY OR SUFFICIENT FOR BEING A LSSP

Let us first recall Example 1.

**Example' 1.** *Let us consider* $\mathcal{L}_G$ *as a hyperbolic paraboloid (a.k.a., saddle point function) centered in* $(1, 1)$ *where* $(1, \varphi)$ *is the principal descent direction and* $(-\varphi, 1)$ *is the principal ascent direction, while* $\mathcal{L}_D$ *is a simple bilinear objective.*

$$\mathcal{L}_G(\theta_1, \theta_2, \varphi) = (\theta_2 - \varphi\theta_1 - 1)^2 - \tfrac{1}{2}(\theta_1 + \varphi\theta_2 - 1)^2, \quad \mathcal{L}_D(\theta_1, \theta_2, \varphi) = \varphi(5\theta_1 + 4\theta_2 - 9)$$

We want to show that $(1, 1, 0)$ is a locally stable stationary point.

*Proof.* The game vector field has the following form,

$$v(\theta_1, \theta_2, \varphi) = \begin{pmatrix} (2\varphi^2 - 1)\theta_1 - 3\varphi\theta_2 + 2\varphi + 1 \\ (2 - \varphi^2)\theta_2 - 3\varphi\theta_1 - 2 + \varphi \\ 5\theta_1 + 4\theta_2 - 9 \end{pmatrix} \tag{15}$$

Thus, $(\theta_1^*, \theta_2^*, \varphi^*) := (1, 1, 0)$ is a stationary point (i.e., $v(\theta_1^*, \theta_2^*, \varphi^*) = 0$). The Jacobian of the game vector field is

$$\nabla v(\theta_1, \theta_2, \varphi) = \begin{pmatrix} 2\varphi^2 - 1 & -3\varphi & 2 - 3\theta_2 \\ -3\varphi & 2 - \varphi^2 & 1 - 3\theta_1 \\ 5 & 4 & 0 \end{pmatrix}, \tag{16}$$

and thus,

$$\nabla v(\theta_1^*, \theta_2^*, \varphi^*) = \begin{pmatrix} -1 & 0 & -1 \\ 0 & 2 & -2 \\ 5 & 4 & 0 \end{pmatrix}. \tag{17}$$

We can verify that the eigenvalues of this matrix have a positive real part with any solver (the eigenvalues of a $3 \times 3$ always have a closed form). For completeness we provide a proof without using the closed form of the eigenvalues. The eigenvalues $\nabla v(\theta_1^*, \theta_2^*, \varphi^*)$ are given by the roots of its characteristic polynomial,

$$\chi(X) := \begin{vmatrix} X + 1 & 0 & 1 \\ 0 & X - 2 & 2 \\ -5 & -4 & 0 \end{vmatrix} = X^3 - X^2 + 11X - 2. \tag{18}$$

This polynomial has a real root in $(0, 1)$ because $\chi(0) = -2 < 0 < 9 = \chi(1)$. Thus we know that, there exists $\alpha \in (0, 1)$ such that,

$$X^3 - X^2 + 11X - 2 = (X - \alpha)(X - \lambda_1)(X - \lambda_2). \tag{19}$$

Then we have the equalities,

$$\alpha \lambda_1 \lambda_2 = 2 \tag{20}$$
$$\alpha + \lambda_1 + \lambda_2 = 1. \tag{21}$$

Thus, since $0 < \alpha < 1$, we have that,

- If $\lambda_1$ and $\lambda_2$ are real, they have the same sign $\lambda_1 \lambda_2 = 2/\alpha > 0$) and thus are positive ($\lambda_1 + \lambda_2 = 1 - \alpha > 0$).

- If $\lambda_1$ is complex then $\lambda_2 = \bar{\lambda}_1$ and thus, $2\Re(\lambda_1) = \lambda_1 + \lambda_2 = 1 - \alpha > 0$.

$\square$

Example 1 showed that LSSP did not imply DNE. Let us construct an example where a game have a DNE which is not locally stable.

**Example 2.** *Consider the non-zero-sum game with the following respective losses for each player,*

$$\mathcal{L}_1(\theta, \phi) = 4\theta^2 + (\tfrac{1}{2}\phi^2 - 1) \cdot \theta \quad and \quad \mathcal{L}_2(\theta, \phi) = (4\theta - 1)\phi + \tfrac{1}{6}\theta^3 \tag{22}$$

This game has two stationary points for $\theta = 0$ and $\phi = \pm 1$. The Jacobian of the dynamics at these two points are

$$\nabla v(0, 1) = \begin{pmatrix} 1 & 1/2 \\ 2 & 1/2 \end{pmatrix} \quad and \quad \nabla v(0, -1) = \begin{pmatrix} 1 & -1/2 \\ 2 & -1/2 \end{pmatrix} \tag{23}$$

Thus,

- The stationary point $(0, 1)$ is a DNE but $\mathrm{Sp}(\nabla v(0, 1)) = \{\frac{3\pm\sqrt{17}}{4}\}$ contains an eigenvalue with negative real part and so is *not* a LSSP.

- The stationary point $(0, -1)$ is *not* a DNE but $\mathrm{Sp}(\nabla v(0, 1)) = \{\frac{1\pm i\sqrt{7}}{4}\}$ contains only eigenvalue with positive real part and so is a LSSP.

## B  Computation of the top-k Eigenvalues of the Jacobian

Neural networks usually have a large number of parameters, this usually makes the storing of the full Jacobian matrix impossible. However the Jacobian vector product can be efficiently computed by using the trick from (Pearlmutter, 1994). Indeed it's easy to show that $\nabla v(\omega) u = \nabla(v(\omega)^T u)$.

To compute the eigenvalues of the Jacobian of the Game, we first compute the gradient $v(\omega)$ over a subset of the dataset. We then define a function that computes the Jacobian vector product using automatic differentiation. We can then use this function to compute the top-k eigenvalues of the Jacobian using the `sparse.linalg.eigs` functions of the Scipy library.

## C  Experimental Details

### C.1  Mixture of Gaussian Experiment

**Dataset.** The Mixture of Gaussian dataset is composed of 10,000 points sampled independently from the following distribution $p_{\mathcal{D}}(x) = \frac{1}{2}\mathcal{N}(2, 0.5) + \frac{1}{2}\mathcal{N}(-2, 1)$ where $\mathcal{N}(\mu, \sigma^2)$ is the probability density function of a 1D-Gaussian distribution with mean $\mu$ and variance $\sigma^2$. The latent variables $z \in \mathbb{R}^d$ are sampled from a standard Normal distribution $\mathcal{N}(0, I_d)$. Because we want to use full-batch methods, we sample 10,000 points that we re-use for each iteration during training.

**Neural Networks Architecture.** Both the generator and discriminator are one hidden layer neural networks with 100 hidden units and ReLU activations.

**WGAN Clipping.** Because of the clipping of the discriminator parameters some components of the gradient of the discriminator's gradient should no be taken into account. In order to compute the relevant path angle we apply the following filter to the gradient:

$$\mathbf{1}\left\{(|\varphi| = \mathbf{c}) \text{ and } (\text{sign}\nabla_\varphi \mathcal{L}_\mathbf{D}(\omega) = -\text{sign}\varphi)\right\} \tag{24}$$

where $\varphi$ is clipped between $-c$ and $c$. If this condition holds for a coordinate of the gradient then it mean that after a gradient step followed by a clipping the value of the coordinate will not change.

| Hyperparameters for WGAN-GP on MoG | |
| --- | --- |
| Batch size | $= 10,000$ (Full-Batch) |
| Number of iterations | $= 30,000$ |
| Learning rate for generator | $= 1 \times 10^{-2}$ |
| Learning rate for discriminator | $= 1 \times 10^{-1}$ |
| Gradient Penalty coefficient | $= 1 \times 10^{-3}$ |

| Hyperparameters for NSGAN on MoG | |
| --- | --- |
| Batch size | $= 10,000$ (Full-Batch) |
| Number of iterations | $= 30,000$ |
| Learning rate for generator | $= 1 \times 10^{-1}$ |
| Learning rate for discriminator | $= 1 \times 10^{-1}$ |

### C.2  MNIST Experiment

**Dataset** We use the training part of MNIST dataset LeCun et al. (2010) (50K examples) for training our models, and scale each image to the range $[-1, 1]$.

**Architecture** We use the DCGAN architecture Radford et al. (2016) for our generator and discriminator, with both the NSGAN and WGAN-GP objectives. The only change we make is that we replace the Batch-norm layer in the discriminator with a Spectral-norm layer Miyato et al. (2018), which we find to stabilize training.

**Training Details**

| **Hyperparameters for NSGAN with Adam** | |
|---|---|
| Batch size | $= 100$ |
| Number of iterations | $= 100,000$ |
| Learning rate for generator | $= 2 \times 10^{-4}$ |
| Learning rate for discriminator | $= 5 \times 10^{-5}$ |
| $\beta_1$ | $= 0.5$ |

| **Hyperparameters for NSGAN with ExtraAdam** | |
|---|---|
| Batch size | $= 100$ |
| Number of iterations | $= 100,000$ |
| Learning rate for generator | $= 2 \times 10^{-4}$ |
| Learning rate for discriminator | $= 5 \times 10^{-5}$ |
| $\beta_1$ | $= 0.9$ |

| **Hyperparameters for WGAN-GP with Adam** | |
|---|---|
| Batch size | $= 100$ |
| Number of iterations | $= 200,000$ |
| Learning rate for generator | $= 8.6 \times 10^{-5}$ |
| Learning rate for discriminator | $= 8.6 \times 10^{-5}$ |
| $\beta_1$ | $= 0.5$ |
| Gradient penalty $\lambda$ | $= 10$ |
| Critic per Gen. iterations $\lambda$ | $= 5$ |

| **Hyperparameters for WGAN-GP with ExtraAdam** | |
|---|---|
| Batch size | $= 100$ |
| Number of iterations | $= 200,000$ |
| Learning rate for generator | $= 8.6 \times 10^{-5}$ |
| Learning rate for discriminator | $= 8.6 \times 10^{-5}$ |
| $\beta_1$ | $= 0.9$ |
| Gradient penalty $\lambda$ | $= 10$ |
| Critic per Gen. iterations $\lambda$ | $= 5$ |

**Computing Inception Score on MNIST**  We compute the inception score (IS) for our models using a LeNet classifier pretrained on MNIST. The average IS score of real MNIST data is 9.9.

### C.3   PATH-ANGLE PLOT

We use the path-angle plot to illustrate the dynamics close to a LSSP. To compute this plot, we need to choose an initial point $\omega$ and an end point $\omega'$. We choose the $\omega$ to be the parameters at initialization, but $\omega'$ can more subtle to choose. In practice, when we use stochastic gradient methods we typically reach a neighborhood of a LSSP where the norm of the gradient is small. However, due to the stochastic noise, we keep moving around the LSSP. In order to be robust to the choice of the end point $\omega'$, we take multiple close-by points during training that have good performance (e.g., high IS in MNIST). In all of figures, we compute the path-angle (and path-norm) for all these end points (with the same start point), and we plot the median path-angle (middle line) and interquartile range (shaded area).

### C.4   INSTABILITY OF GRADIENT DESCENT

For the MoG dataset we tried both the extragradient method (Korpelevich, 1976; Gidel et al., 2019a) and the standard gradient descent. We observed that gradient descent leads to unstable results. In

particular the norm of the gradient has very large variance compared to extragradient this is shown in Fig. 7.

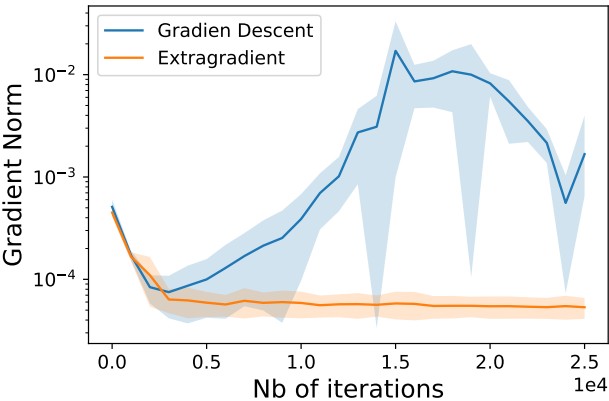

Figure 7: The norm of gradient during training for the standard GAN objective. We observe that while extra-gradient reaches low norm which indicates that it has converged, the gradient descent on the contrary doesn't seem to converge.

## C.5 ADDITIONAL RESULTS WITH ADAM

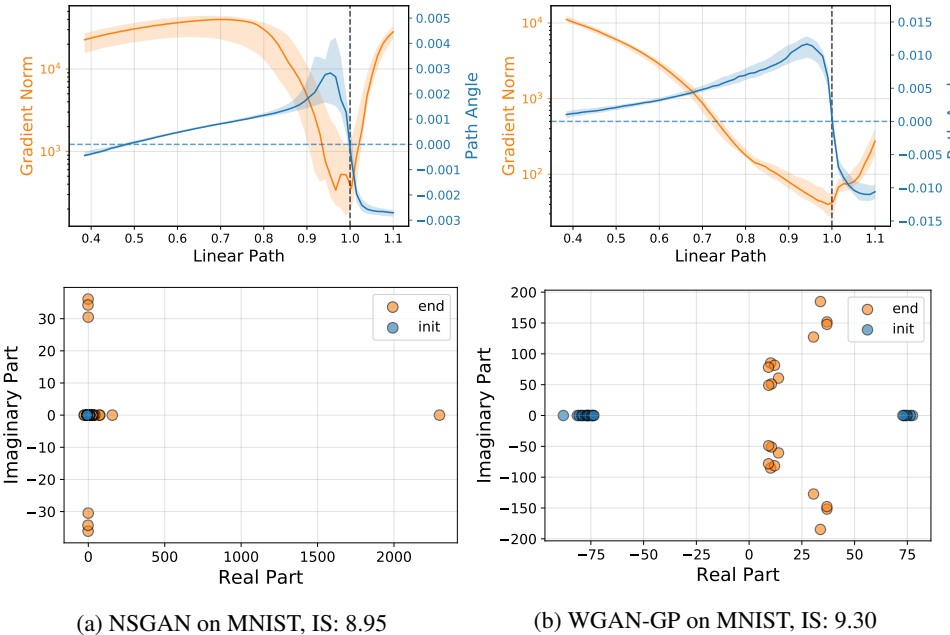

(a) NSGAN on MNIST, IS: 8.95

(b) WGAN-GP on MNIST, IS: 9.30

Figure 8: Path-angle and Eigenvalues computed on MNIST with Adam.

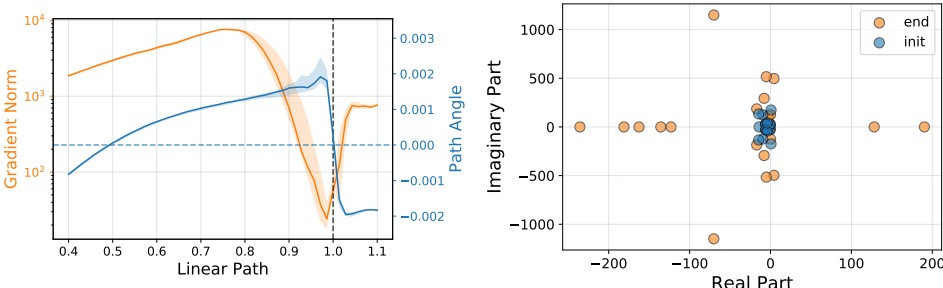

Figure 9: Path-angle and Eigenvalues for NSGAN on CIFAR10 computed on CIFAR10 with Adam. We can see that the model has eigenvalues with negative real part, this means that we've actually reached an unstable point.

