# OpenReview forum: "A Closer Look at the Optimization Landscapes of Generative Adversarial Networks"
_ICLR.cc/2020/Conference — Accept (Poster)_

### Official Review · AnonReviewer3 · 2019-10-16
**Official Blind Review #3**

**Rating:** 6

**Review:**

This paper tries to provide a deeper understanding of the training dynamics of GANs in practice via characterizing and visualizing the rotation and attraction phenomena nearby a locally stable stationary point (LSSP) and questions the necessity to access a differential/local Nash equilibrium (LNE). In particular, this paper first discusses the difference between LSSP and LNE and formalize the notions of rotation and attraction around LSSP in games. Then, this paper proposes the path angle to visualize the rotation and attraction nearby an LSSP. The path angle is a function that maps linearly distributed points in the line, which is determined by an initial parameter set and a well-trained parameter set, to the angles between the line and the gradient of a given point in that line. The rotation and attraction phenomena can be observed in the plot of the path angle as  "a quick sign switch" and "a bump" nearby 1, respectively. The experiments empirically demonstrate that: 1. rotation exists in the training dynamics of practical GANs; 2. GANs often converge to an LSSP than an LNE, but still, achieve good results.

Generally, this paper is interesting and well-written. The contribution is clearly presented and the literature is well discussed. However, I have some questions to be clarified by the authors as follows.

1. In Sec 3.2., this paper tries to motivate the readers to notice the difference between the LSSP and DNE by introducing Example 1. However, I notice that there is a gap that hasn't been presented clearly: Example 1 is a general game but does not correspond to a GAN, which is of the most interest in the paper. Besides, the generator loss at the optimum should be (theta_2 - 1)^2 - 1/2(theta_1 - 1)^2 instead of theta_2^2 - 1/2theta_1^2.

2. For the rotation around LSSP, existing work, including Mescheder et al. (2018), Gidel et al. (2019b),  has a prior discussion. Besides, it is intuitive that an LSSP is not an LNE in practical GANs with high probability because finding a descent direction is easy given such a high-dimensional space. It is also possible to find a sharp descent direction nearby an LSSP because the norm of the gradient is averaged across all dimensions. It is good to formulate these observations in a precise way but it would be better to see further implications of the two observations. If so, the paper quality will be significantly improved.

3. A minor thing is why (c) and (f) in Figure 3 and Figure 4 use different metrics, i.e. FID and IS, respectively?

I also note that this paper has 10 pages and should be expected at a higher level than other accepted papers. Given all these conditions, I think I make it clear why I give a rating 6 currently.

By the way, I'm not absolutely confident about the comments because I didn't work on analyzing the dynamics of GANs. I'll appreciate it if my issues can be addressed or a potential misunderstanding can be corrected.

**Experience Assessment:**

I have read many papers in this area.

**Review Assessment: Checking Correctness Of Derivations And Theory:**

I assessed the sensibility of the derivations and theory.

**Review Assessment: Checking Correctness Of Experiments:**

I assessed the sensibility of the experiments.

**Review Assessment: Thoroughness In Paper Reading:**

I read the paper at least twice and used my best judgement in assessing the paper.

---

> ### Author Response · Authors · 2019-11-12
> **Response to reviewer 3**
>
> We first want to thank the reviewer for his positive feedback and useful comments. We tried to address as well as possible his questions:
>
> Q1: “In Sec 3.2., this paper tries to motivate the readers to notice the difference between the LSSP and DNE by introducing Example 1. However, I notice that there is a gap that hasn't been presented clearly: Example 1 is a general game but does not correspond to a GAN, which is of the most interest in the paper”.
>
> A1: We agree that formally Example 1 might not be a GAN (some two-player games cannot be cast as GANs). The goal of this simple example was to give intuition on how an LSSP may also be a saddle point for the generator loss. We argue that this example is closely related to practical GANs since it gives insights on the phenomenon of non-Nash stable attractors practically observed in Section 5.2.
>
> Q2: “For the rotation around LSSP, existing work, including Mescheder et al. (2018), Gidel et al. (2019b),  has a prior discussion. Besides, it is intuitive that an LSSP is not an LNE in practical GANs with high probability because finding a descent direction is easy given such a high-dimensional space. It is also possible to find a sharp descent direction nearby an LSSP because the norm of the gradient is averaged across all dimensions.”
>
> A2: We agree with R3, that with the right practical perspective it is actually quite intuitive that LSSP are not LNE in GANs. Nevertheless, we want to insist on two points, first this is not necessarily the current consensus in the literature, in particular the assumptions used in theory to show GAN convergence are not compatible with this observations and are thus may not represent what happens in practice (see for example assumptions in [Nagarajan and Kolter (2017); Mescheder et al. (2018)] where the generator is assumed to be able to roughly capture the real data distribution), second to our knowledge we’re the first to clearly show that this intuitive phenomenon actually happens in GANs.
>
> We believe that GANs do not converge to local Nash equilibria because standard neural networks are quite sensitive to adversarial examples, thus it is easy for the generator to find a descent direction that “fools” the discriminator. Understanding precisely this phenomenon is an interesting potential direction of research and we leave it for future work.
>
> 3. A minor thing is why (c) and (f) in Figure 3 and Figure 4 use different metrics, i.e. FID and IS, respectively?
>
> The purpose of indicating a performance metric was just to show that the models we were dealing with had satisfying performance in terms of standard metrics. The different model we used came from different code base and didn’t use the same metrics. We computed the Inception Score for the NSGAN models and will add it to the paper.

---

> > ### Comment · AnonReviewer3 · 2019-11-14
> > **Response to the author feedback**
> >
> > Thanks for your clarification.

---

> > > ### Author Response · Authors · 2019-11-14
> > > **Updated Paper**
> > >
> > > We updated the paper replacing the FID score for NSGAN with the Inception Score (as mentioned in our answer).

---

### Official Review · AnonReviewer1 · 2019-10-21
**Official Blind Review #1**

**Rating:** 6

**Review:**

Summary:

This paper proposes visualization techniques for the optimization landscape in GANs. The primary tool presented in this paper is a quantity called path-angle, which looks at the angle between the game vector field and the linear path between a point away from a stationary point and a point near a stationary point. The paper present examples of the visualization for dynamics with pure attraction, pure rotation, and a mix of attraction and rotation. Along with this, the authors propose to look at the eigenvalues of the game Jacobian and the individual player Hessian’s to evaluate convergence in GANs. The paper presents application of the tools on GANs trained with NSGAN and WGAN-GP objectives on a mixture of Gaussians, MNIST, and CIFAR10. The primary observation is that the generator performance is good, but the algorithms converge to non-Nash stable attractors. Moreover, it is shown using the path-angle plots that GANs exhibit rotational behavior around stable points.

Review:

There has been a lot of work in the past few years (and ongoing) on principled training approaches for GANs. The objective of the algorithms is typically to converge a differential Nash equilibrium and/or to reach a stable point of the dynamics quickly. In my view, this work fills some of the gap on the empirical side of things with respect to each goal.

Notably, a main idea to speed up convergence in GANs is to change the gradient play dynamics so rotational components are neutralized. The path angle visualization provides a novel tool to evaluate the empirical ability of any dynamics proposed for GANs to cancel out rotational components. Since it is generally known that gradient play dynamics are susceptible to cycling, I would have been interesting in seeing the path angle plots for some recently proposed algorithms such as consensus, symplectic gradient adjustment, stable opponent shaping, local symplectic surgery, etc to see how they compare. This would have made the experiments using the path angle visualization stronger in my view. Nonetheless, the path angle tool is useful and I can foresee it being commonly used in the future.

Aside from neutralizing rotational components, dynamics have been proposed with the goal of avoiding non-Nash stable attractors and converging only to differential Nash equilibria. However, to my knowledge, there has not been much, if any, evaluation in GANs to see if the methods are in fact converging to Nash equilibria as theory may predict. While simple, I found it interesting to evaluate the eigenvalues of the relevant quantities at convergence. I am curious why the authors evaluate the top-k eigenvalues in terms of magnitude? The scipy package referenced in the appendix can compute the largest and smallest real eigenvalues, which is what it seems like you would want to evaluate the definiteness of the game Jacobian and the individual player Hessians. The most interesting empirical result in the paper to me was that it is common to converge to non-Nash stable attractors using standard training techniques and at such stable points the generator performance is strong. This is an important observation and  may cause some consideration of what points should be sought in GANs. I am not fully convinced this is always what the dynamics would always converge to depending on the network, learning rates, optimization methods, etc, but showing that it can be the case is useful.

Overall, I think this paper introduces some useful tools to interpret the performance in GANs and to help understand the behavior of training dynamics. The main tool introduced was the path angle visualization and the primary empirical result was that standard GAN methods may reach non-Nash stable attractors and perform well. The paper probably be condensed in the first 4 pages, so that more experimental results could be presented and this would make the paper stronger.

Post Response: Thanks for the response. I believe this paper should be accepted.

**Experience Assessment:**

I have published one or two papers in this area.

**Review Assessment: Checking Correctness Of Derivations And Theory:**

I assessed the sensibility of the derivations and theory.

**Review Assessment: Checking Correctness Of Experiments:**

I assessed the sensibility of the experiments.

**Review Assessment: Thoroughness In Paper Reading:**

I read the paper thoroughly.

---

> ### Author Response · Authors · 2019-11-12
> **Response to reviewer 1**
>
> We would like to thank R1 for the positive feedback. We share the belief that the visualization techniques in particular the path-angle method can be useful tools in the future in order to bridge part of the gap between theory and practice. We also want to address a few points made by the reviewer:
>
> 1) Regarding plotting path angle for some recently proposed algorithms such as consensus and symplectic gradient adjustment, we would like to first emphasize that we tried to focus on the standard training setup of GANs, i.e. using Adam as the optimization method.  In addition, we used ExtraAdam (Gidel et al. 2019), which was recently proposed and is supposed to handle better the rotational behavior of games. We didn’t find any significant difference between the two different methods, apart from ExtraAdam being, in general, more stable than Adam. We will make the code publicly available, and hope people will use it to compare their methods and how they handle rotations.
>
> More importantly, as detailed in the answer for R4 on “Why does interpolating between initialization and the final learned weight vector tell us about the rotational dynamics in high dimensions”, our goal was not to do an exhaustive comparison of the properties of the point found by a variety of training methods: even though each optimization will lead to a different optimization trajectory (and thus a different solution) we wanted to focus primarily on the constitutive properties of the vector field in a way agnostic to the choice of the optimization method.
>
>
> 2) Regarding the questions: ‘should we look at the top-k eigenvalues in terms of magnitude or should we compute the largest and smallest eigenvalues ?’
> We think that both are meaningful and could be used. However it is usually more expensive to compute the smallest eigenvalues than the largest (please refer to Alain et al 2019). Therefore, for the same computational budget, we can compute more eigenvalues, which means that we have more information about the spectrum of the Jacobian. This is the main reason why we chose to focus on the top-k eigenvalues. Secondly, we believe that the largest eigenvalues in terms of magnitude are the ones that locally set the main behavior of the dynamics.

---

### Official Review · AnonReviewer4 · 2019-10-22
**Official Blind Review #4**

**Rating:** 6

**Review:**

The authors present a study of GAN dynamics in training with the goal of understanding whether rotational behavior occurs when training GANs on real world datasets, and whether training methods find local Nash equilibria.


The authors start by motivating the study of a game vector field with a toy example in which we can see all relevant behavior in the relevant directions. The work investigates the game vector field with a visualization technique called “path-angle” that attempts to alleviate the problem of high dimensionality by only looking at the cosine similarity (between the linear interpolation between the two points versus the true gradient at the point) along a path between two (concatenated) weight vectors at a time. The work also investigates by looking at the gradient norms of weights in these optimization trajectories.


The work uses these techniques to visualize the dynamics of GANs trained on standard datasets. The authors find that GANs do not converge to local Nash equilibria, that each player ends at a saddle point, and state evidence for “rotational behavior” in GAN dynamics.


The finding that GAN training methods do not find local Nash equilibria is interesting. However, it is unclear to me what the experiments presented show about GAN training dynamics (in particular, it is unclear to me what rotational dynamics are in the context of GANs and what consequences they have for training). I have listed more detailed feedback below.


Detailed feedback:
* Section 3.1: What is the formulation of Mescheder et al 2017? This should be explicitly stated in the paper.
* The authors never explicitly, formally define what it means for there to be rotational behavior. However, this terminology is used frequently throughout the paper, particularly in the empirical section (5.1). What does rotational component mean in this Section?
* What is the motivation for completing the path based landscape visualization methods between a random initialization and the final weight vector? Is there a reason why the actual iterates were not investigated with this method?
* Why does the bump in Figure 3 imply that there is a non zero rotational component?
* It would be good to complete a more thorough understanding of the spectra of hessians at convergence; the extent of the experiments in Figure 5 and 6 appears to just be 3 training runs; a larger sample size would be good to establish trends.
* What is the motivation for visualizing via the path based methods? Why does interpolating between initialization and the final learned weight vector tell us about the rotational dynamics in high dimensions? This line may not be representative of the optimization trajectory followed by the actual iterates.

**Experience Assessment:**

I do not know much about this area.

**Review Assessment: Checking Correctness Of Derivations And Theory:**

I did not assess the derivations or theory.

**Review Assessment: Checking Correctness Of Experiments:**

I assessed the sensibility of the experiments.

**Review Assessment: Thoroughness In Paper Reading:**

I read the paper at least twice and used my best judgement in assessing the paper.

---

> ### Author Response · Authors · 2019-11-12
> **Response to reviewer 4 (1/2)**
>
> While R4 acknowledged our contribution showing that GANs do not converge to Nash equilibria, we believe the reviewer has missed the other part of our contributions about rotations and how we can use the proposed path-angle to detect such rotations. We answered the different questions raised by R4, hoping it will clarify the discussion around rotations and make our contributions about the path angle more clear.
>
>
> Q1: Section 3.1: What is the formulation of Mescheder et al 2017? This should be explicitly stated in the paper.
>
> A1: What we mean in section 3.1 is that the GAN formulation we use is similar to the formulation of Mescheder et al 2017, which has introduced it for a general definition of two-player games but is not limited to GANs. We have decided to keep this general notation as we think the visualization tools we describe in our paper could be of interest to a wider audience than just the GAN community.
>
> More precisely, equation (1) in our paper is the same as equation (4) in Mescheder et al 2017. This equation is the definition of a Nash-Equilibrium for a general two-player game. In the case of GANs, we have a Generator $G_\theta$ with parameters $\theta$ that is trying to solve $\displaystyle min_{\theta} L_G(\theta,\phi)$ and a discriminator $D_\phi$ with parameters $\phi$ which  is trying to solve $\displaystyle min_{\phi} L_D(\theta,\phi)$. The reader can see that this is quite a general formulation since we can replace $L_D$and $L_G$ by any kind of loss that has been proposed in the literature.
>
> As an example, if we consider the following losses for the Generator and Discriminator: $L_G(\theta,\phi) = \mathbb{E}_{z\sim P_z}[\log(1-D_\phi(G_\theta(z)))]$ and $L_D(\theta,\phi) = -\mathbb{E}_{x\sim P_{data}}[\log D_\phi(x)] - \mathbb{E}_{z\sim P_z}[\log(1-D_\phi(G_\theta(z)))]$ then solving equation (1) is equivalent to solving the original GAN formulation of Goodfellow et al 2014. We refer the reviewer to [3] for a more in depth presentation of how other GAN variants can be formulated as two-player games between the generator and discriminator.
>
>
> Q2: The authors never explicitly, formally define what it means for there to be rotational behavior. However, this terminology is used frequently throughout the paper, particularly in the empirical section (5.1). What does rotational component mean in this Section?
>
> A2: Please note that the notion of rotation is formally defined in section 3.3 with Proposition 1 and further explained in the following paragraph. We summarize this in the following points:
> 1. Close to a stationary point the dynamics of the joint state $\omega$ (i.e., the concatenation of the parameters of the generator and the discriminator) can be decomposed into components that either are attracted to or rotate around (or both) the stationary point. The rotation of the dynamics of these components is explicitly given by a two dimensional rotation matrix $\begin{pmatrix} \cos \theta & \sin \theta \\ -\cos \theta & \sin \theta \end{pmatrix}$ (see Case 2 and 3 of Proposition 1).
> 2. Proposition 1 emphasizes that imaginary part of the eigenvalues of the Jacobian $\nabla v(\omega^*)$ induce rotations for some components of the dynamics of $\omega$.
> 3. The rotational aspect of the dynamics of $\omega$ has several interpretations:
> - The iterates come back to a neighborhood of the initial point (see for instance [1,2] for more details).
> - The gradient is orthogonal to the direction toward the optimal solution (see e.g., Fig 2b and 2c). Our path angle
>              method leverages this characterization (detailed in Section 4.3) in order to detect rotations without computing
>              eigenvalues.
> 4. In Section 5.1 we present two methods for detecting rotations in practice: the direct computation of the imaginary part of the eigenvalues of the Jacobian and the ‘bump’ in the path angles. In Section 4.3, we explain why bumps only appear on the path angle when the eigenvalues of $\nabla v(\omega^*)$ have some non-zero imaginary parts.
>
> Q4: Why does the bump in Figure 3 imply that there is a non zero rotational component?
>
> A4: We explain in Figure 2 and Section 4.3 how to relate Proposition 1 to practical insights, particularly, why a bump implies a non-zero rotational component. The idea is the following: if there is a rotation around the equilibrium (see fig 2b)), the vector field is going to point to the right on one side of the linear path and to the left on the other side (because the center of the rotation is at the optimum at the middle of the path). Since the vector field is continuous and goes from the right side to the left side of the path it has to be perfectly aligned with the linear path in between (thus a cosine similarity of +/- 1).

---

> > ### Author Response · Authors · 2019-11-12
> > **Response to reviewer 4 (2/2)**
> >
> > Q3 & Q6: What is the motivation for completing the path based landscape visualization methods between a random initialization and the final weight vector? Is there a reason why the actual iterates were not investigated with this method?
> >
> > What is the motivation for visualizing via the path based methods? Why does interpolating between initialization and the final learned weight vector tell us about the rotational dynamics in high dimensions? This line may not be representative of the optimization trajectory followed by the actual iterates.
> >
> > A3 & A6: First we want to mention that in the context of single objective optimization people have already looked at a linear path between initialization and the final weight vector to get insight on the optimization landscape of deep neural networks see for example (Goodfellow et al 2015). While looking at a linear path has its own limitations, we believe this is a necessary first step to get insight into the optimization landscape of GANs.
> >
> > We actually also considered plotting the path-angle along the non-linear path defined by the actual iterates without success. In particular, the linear path is much easier to understand and interpret than the path defined by the actual iterates, which is highly non-linear. In particular, we have a nice interpretation of what the path-angle should look like on the archetypal examples (described in figure 2), but it is very hard to postulate what the path-angle for a non-linear path would look like in general. We actually confirmed this experimentally by plotting the path-angle along the path defined by the actual iterates, and found them to be noisy and hard to interpret.
> >
> > Finally, an important point to mention is that we were not actually interested in the trajectory followed by the method used, but we were rather interested in quantifying the amount of rotation in the game vector field itself independent of the training method used. Since the linear path only depends on the final iterate, it gives us information about the geometry of the landscape, that is less dependent on the optimization method used. However an important point to mention is that different optimization methods may find different final iterates with different properties. In our experiments we didn’t find any significant difference between different methods, except that some (i.e. Adam) are in general less stable than others (i.e. ExtraAdam).
> > We believe that looking at the trajectory of specific algorithms might bring other useful information about the training dynamics, but we think this is out of the scope of the paper and leave this for future work.
> >
> >
> > Q5: It would be good to complete a more thorough understanding of the spectra of hessians at convergence; the extent of the experiments in Figure 5 and 6 appears to just be 3 training runs; a larger sample size would be good to establish trends.
> >
> > A5: In total we have run several experiments on: 3 objectives (WGAN, WGAN-GP, NSGAN), 2 optimization methods (Adam, ExtraAdam), 3 datasets (MoG, MNIST, CIFAR10), where each time we tried different hyperparameters and selected the parameters giving sartisfying performance. We also used different seeds, and the observations were consistent across models, in the paper we only focus on a few models for clarity. Particularly, we think that R4 missed that we provide in the appendix similar experiments as the one presented in the main paper with another optimization method (Adam instead of ExtraAdam).
> >
> >
> > Citations:
> > [1] Mertikopoulos, Panayotis, Christos Papadimitriou, and Georgios Piliouras. "Cycles in adversarial regularized learning." Proceedings of the Twenty-Ninth Annual ACM-SIAM Symposium on Discrete Algorithms. Society for Industrial and Applied Mathematics, 2018.
> > [2] Bailey, James P., Gauthier Gidel, and Georgios Piliouras. "Finite Regret and Cycles with Fixed Step-Size via Alternating Gradient Descent-Ascent." arXiv preprint arXiv:1907.04392 (2019).
> > [3] Fedus, W., Rosca, M., Lakshminarayanan, B., Dai, A. M., Mohamed, S., & Goodfellow, I. “Many Paths to Equilibrium: GANs Do Not Need to Decrease a Divergence At Every Step.” International Conference on Learning Representations, 2018.

---

> > > ### Comment · AnonReviewer4 · 2019-11-14
> > > **Thank you**
> > >
> > > Thank you for the clarification and thank you for answering my questions clearly!

---

### Decision · Program_Chairs · 2019-12-19

**Decision:**

Accept (Poster)

**Comment:**

This is an interesting contribution that sheds some light on a well-studied but still poorly understood problem. I think it might be of interest to the community.